# Scan Order in Gibbs Sampling: Models in Which it Matters and Bounds on How Much

**Bryan He, Christopher De Sa, Ioannis Mitliagkas, and Christopher Ré**
Stanford University
{bryanhe,cdesa,imit,chrismre}@stanford.edu

## Abstract

Gibbs sampling is a Markov Chain Monte Carlo sampling technique that iteratively samples variables from their conditional distributions. There are two common scan orders for the variables: random scan and systematic scan. Due to the benefits of locality in hardware, systematic scan is commonly used, even though most statistical guarantees are only for random scan. While it has been conjectured that the mixing times of random scan and systematic scan do not differ by more than a logarithmic factor, we show by counterexample that this is not the case, and we prove that that the mixing times do not differ by more than a polynomial factor under mild conditions. To prove these relative bounds, we introduce a method of augmenting the state space to study systematic scan using conductance.

## 1 Introduction

Gibbs sampling, or Glauber dynamics, is a Markov chain Monte Carlo method that draws approximate samples from multivariate distributions that are difficult to sample directly [9; 15, p. 40]. A major use of Gibbs sampling is marginal inference: the estimation of the marginal distributions of some variables of interest [8]. Some applications include various computer vision tasks [9, 23, 24], information extraction [7], and latent Dirichlet allocation for topic modeling [11]. Gibbs sampling is simple to implement and quickly produces accurate samples for many models, so it is widely used and available in popular libraries such as OpenBUGS [16], FACTORIE [17], JAGS [18], and MADlib [14].

Gibbs sampling (Algorithm 1) iteratively selects a single variable and resamples it from its conditional distribution, given the other variables in the model. The method that selects the variable index to sample ($s$ in Algorithm 1) is called the *scan order*. Two scan orders are commonly used: random scan and systematic scan (also known as deterministic or sequential scan). In random scan, the variable to sample is selected uniformly and independently at random at each iteration. In systematic scan, a fixed permutation is selected, and the variables are repeatedly selected in that order. The existence of these two distinct options raises an obvious question—which scan order produces accurate samples more quickly? This question has two components: hardware efficiency (how long does each iteration take?) and statistical efficiency (how many iterations are needed to produce an accurate sample?).

From the hardware efficiency perspective, systematic scans are clearly superior [21, 22]. Systematic scans have good spatial locality because they access the variables in linear order, which makes their iterations run faster on hardware. As a result, systematic scans are commonly used in practice.

Comparing the two scan orders is much more interesting from the perspective of statistical efficiency, which we focus on for the rest of this paper. Statistical efficiency is measured by the *mixing time*, which is the number of iterations needed to obtain an accurate sample [15, p. 55]. The mixing times of random scan and systematic scan have been studied, and there is a longstanding conjecture [3; 15, p. 300] that systematic scan (1) never mixes more than a constant factor slower than random scan and (2) never mixes more than a logarithmic factor faster than random scan. This conjecture implies that the choice of scan order does not have a large effect on performance.

---

**Algorithm 1** Gibbs sampler

---

**input** Variables $x_i$ for $1 \le i \le n$, and target distribution $\pi$
  Initialize $x_1, \ldots, x_n$
  **loop**
    Select variable index $s$ from $\{1, \ldots, n\}$
    Sample $x_s$ from the conditional distribution $\mathbf{P}_\pi \left( X_s \mid X_{\{1,\ldots,n\} \setminus \{s\}} \right)$
  **end loop**

---

Recently, Roberts and Rosenthal [20] described a model in which systematic scan mixes more slowly than random scan by a polynomial factor; this disproves direction (1) of this conjecture. Independently, we constructed other models for which the scan order has a significant effect on mixing time. This raises the question: what are the true bounds on the difference between these mixing times? In this paper, we address this question and make the following contributions.

- In Section 3, we study the effect of the variable permutation chosen for systematic scan on the mixing time. In particular, in Section 3.1, we construct a model for which a systematic scan mixes a polynomial factor faster than random scan, disproving direction (2) of the conjecture, and in Section 3.2, we construct a model for which the systematic scan with the worst-case permutation results in a mixing time that is slower by a polynomial factor than both the best-case systematic scan permutation and random scan.

- In Section 4, we empirically verify the mixing times of the models we construct, and we analyze how the mixing time changes as a function of the permutation.

- In Section 5, we prove a weaker version of the conjecture described above, providing relative bounds on the mixing times of random and systematic scan. Specifically, under mild conditions, different scan orders can only change the mixing time by a polynomial factor. To obtain these bounds, we introduce a method of augmenting the state space of Gibbs sampling so that the method of conductance can be applied to analyze its dynamics.

## 2 Related Work

Recent work has made progress on analyzing the mixing time of Gibbs sampling, but there are still major limitations to our understanding. In particular, most results are only for specific models or for random scan. For example, mixing times are known for Mallow's model [1, 4], and colorings of a graph [5] for both random and systematic scan, but these are not applicable to general models. On the other hand, random scan has been shown to mix in polynomial time for models that satisfy structural conditions – such as having close-to-modular energy functions [10] or having bounded hierarchy width and factor weights [2] – but corresponding results for for systematic scan are not known. The major exception to these limitations is Dobrushin's condition, which guarantees $O(n \log n)$ mixing for both random scan and systematic scan [6, 13]. However, many models of interest with close-to-modular energy functions or bounded hierarchy width do not satisfy Dobrushin's condition.

A similar choice of scan order appears in stochastic gradient descent (SGD), where the standard SGD algorithm uses random scan, and the incremental gradient method (IGM) uses systematic scan. In contrast to Gibbs sampling, avoiding "bad permutations" in the IGM is known to be important to ensure fast convergence [12, 19]. In this paper, we bring some intuition about the existence of bad permutations from SGD to Gibbs sampling.

## 3 Models in Which Scan Order Matters

Despite a lack of theoretical results regarding the effect of scan order on mixing times, it is generally believed that scan order only has a small effect on mixing time. In this section, we first define relevant terms and state some common conjectures regarding scan order. Afterwards, we give several counterexamples showing that the scan order can have asymptotic effects on the mixing time.

The *total variation distance* between two probability distributions $\mu$ and $\nu$ on $\Omega$ is [15, p. 47]

$$\|\mu - \nu\|_{\text{TV}} = \max_{A \subseteq \Omega} |\mu(A) - \nu(A)|.$$

Table 1: Models and Approximate Mixing Times

| Model | $t_{\mathrm{mix}}(R)$ | $\min\limits_{\alpha} t_{\mathrm{mix}}(S_\alpha)$ | $\max\limits_{\alpha} t_{\mathrm{mix}}(S_\alpha)$ |
|---|---|---|---|
| Sequence of Dependencies | $n^2$ | $n$ | $n^2$ |
| Two Islands | $2^n$ | $2^n$ | $n2^n$ |
| Discrete Pyramid [20] | $n$ | $n^3$ | $n^3$ |
| Memorize and Repeat | $n^3$ | $n^2$ | $n^2$ |
| Soft Dependencies | $n^{3/2}$ | $n$ | $n^2$ |

The *mixing time* is the minimum number of steps needed to guarantee that the total variation distance between the true and estimated distributions is below a given threshold $\epsilon$ from any starting distribution. Formally, the *mixing time* of a stochastic process $P$ with transition matrix $P^{(t)}$ after $t$ steps and stationary distribution $\pi$ is [15, p. 55]

$$t_{\mathrm{mix}}(P, \epsilon) = \min\left\{t : \max_{\mu} \|P^{(t)}\mu - \pi\|_{\mathrm{TV}} \le \epsilon\right\},$$

where the maximum is taken over the distribution $\mu$ of the initial state of the process. When comparing the statistical efficiency of systematic scan and random scan, it would be useful to establish, for any systematic scan process $S$ and random scan process $R$ on the same $n$-variable model, a relative bound of the form

$$F_1(\epsilon, n, t_{\mathrm{mix}}(R, \epsilon)) \le t_{\mathrm{mix}}(S, \epsilon) \le F_2(\epsilon, n, t_{\mathrm{mix}}(R, \epsilon)) \tag{1}$$

for some functions $F_1$ and $F_2$. Similarly, to bound the effect that the choice of permutation can have on the mixing time, it would be useful to know, for any two systematic scan processes $S_\alpha$ and $S_\beta$ with different permutations on the same model, that for some function $F_3$,

$$t_{\mathrm{mix}}(S_\alpha, \epsilon) \le F_3(\epsilon, n, t_{\mathrm{mix}}(S_\beta, \epsilon)). \tag{2}$$

Diaconis [3] and Levin et al. [15, p. 300] conjecture that systematic scan is never more than a constant factor slower or a logarithmic factor faster than random scan. This is equivalent to choosing $F_1(\epsilon, n, t) = C_1(\epsilon) \cdot t \cdot (\log n)^{-1}$ and $F_2(\epsilon, n, t) = C_2(\epsilon) \cdot t$ in the inequality in (1), for some functions $C_1$ and $C_2$. It is also commonly believed that all systematic scans mix at the same asymptotic rate, which is equivalent to choosing $F_3(\epsilon, n, t) = C_3(\epsilon) \cdot t$ in (2).

These conjectures imply that using systematic scan instead of random scan will not result in significant consequences, at least asymptotically, and that the particular permutation used for systematic scan is not important. However, we show that neither conjecture is true by constructing models (listed in Table 1) in which the scan order has substantial asymptotic effects on mixing time.

In the rest of this section, we go through two models in detail to highlight the diversity of behaviors that different scan orders can have. First, we construct the *sequence of dependencies* model, for which a single "good permutation" of systematic scan mixes faster, by a polynomial factor, than both random scan and systematic scans using most other permutations. This serves as a counterexample to the conjectured lower bounds (i.e. the choice of $F_1$ and $F_3$) on the mixing time of systematic scan. Second, we construct the *two islands* model, for which a small set of "bad permutations" mix very slowly in comparison to random scan and most other systematic scans. This contradicts the conjectured upper bounds (i.e. the choice of $F_2$ and $F_3$). For completeness, we also discuss the *discrete pyramid* model introduced by Roberts and Rosenthal [20], which contradicts the conjectured choice of $F_2$. Table 1 lists several additional models we constructed: these models further explore the space of asymptotic comparisons among scan orders, but for brevity we defer them to the appendix.

### 3.1 Sequence of Dependencies

The first model we describe is the sequence of dependencies model (Figure 1a), where we explore how fast systematic scan can be by allowing a specific good permutation to mix rapidly. The sequence of dependencies model achieves this by having the property that, at any time, progress towards mixing is only made if a particular variable is sampled; this variable is always the one that is chosen by the good permutation. As a result, while a systematic scan using the good permutation makes progress at

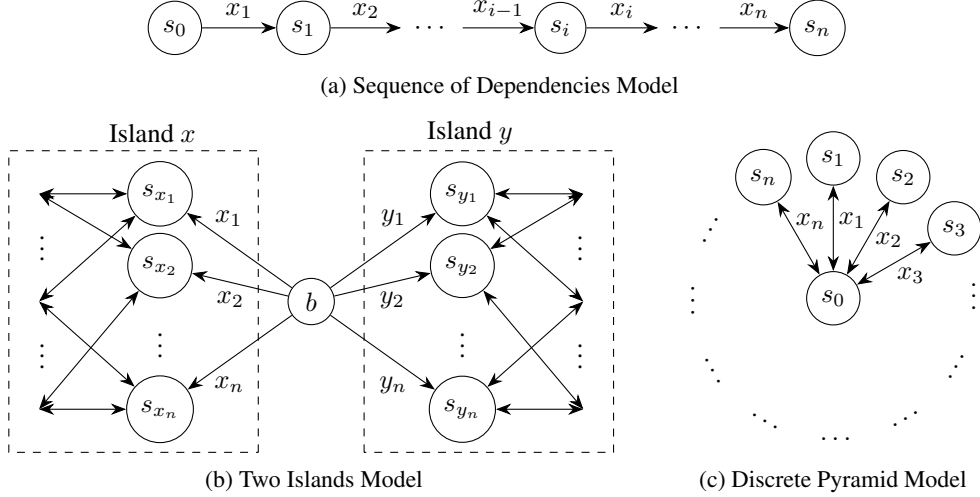

(a) Sequence of Dependencies Model

(b) Two Islands Model

(c) Discrete Pyramid Model

Figure 1: State space of the models.

every step, both random scan and other systematic scans often fail to progress, which leads to a gap between their mixing times. Thus, this model exhibits two surprising behaviors: (1) one systematic scan is polynomially better than random scan and (2) systematic scans using different permutations have polynomial differences in mixing times. We now describe this model in detail.

**Variables** There are $n$ binary variables $x_1, \ldots, x_n$. Independently, each variable has a very strong prior of being true. However, variable $x_i$ is never true unless $x_{i-1}$ is also true. The unnormalized probability distribution is the following, where $M$ is a very large constant.[1]

$$P(x) \propto \begin{cases} 0 & \text{if } x_i \text{ is true and } x_{i-1} \text{ is false for some } i \in \{2, \ldots, n\} \\ M^{\sum_{i=1}^{n} x_i} & \text{otherwise} \end{cases}$$

**State Space** There are $n+1$ states with non-zero probability: $s_0, \ldots, s_n$, where $s_i$ is the state where the first $i$ variables are true and the remaining $n - i$ variables are false. In the stationary distribution, $s_n$ has almost all of the mass due to the strong priors on the variables, so reaching $s_n$ is essentially equivalent to mixing because the total variation distance from the stationary distribution is equal to the mass not on $s_n$. Notice that sampling $x_i$ will almost always move the state from $s_{i-1}$ to $s_i$, very rarely move it from $s_i$ to $s_{i-1}$, and can have no other effect. The worst-case starting state is $s_0$, where the variables must be sampled in the order $x_1, \ldots, x_n$ for this model to mix.

**Random Scan** The number of steps needed to transition from $s_0$ to $s_1$ is distributed as a geometric random variable with mean $n$ (variables are randomly selected, and specifically $x_1$ must be selected). Similarly, the number of steps needed to transition from $s_{i-1}$ to $s_i$ is distributed as a geometric random variable with mean $n$. In total, there are $n$ transitions, so $O(n^2)$ steps are needed to mix.

**Best Systematic Scan** The best systematic scan uses the order $x_1, x_2, \ldots, x_n$. For this scan, one sweep will reach $s_n$ no matter what the starting state is, so the mixing time is $n$.

**Worst Systematic Scan** The worst systematic scan uses the order $x_n, x_{n-1}, \ldots, x_1$. The first sweep only uses $x_1$, the second sweep only uses $x_2$, and in general, any sweep only makes progress using one transition. Finally, in the $n$-th sweep, $x_n$ is used in the first step. Thus, this process mixes in $n(n-1) + 1$ steps, which is $O(n^2)$.

## 3.2 Two Islands

With the sequence of dependencies model, we showed that a single good permutation can mix much faster than other scan orders. Next, we describe the two islands model (Figure 1b), which has the

opposite behavior: it has bad permutations that yield much slower mixing times. The two islands model achieves this by having two disjoint blocks of variables such that consecutively sampling two variables from the same block accomplishes very little. As a result, a systematic scan that uses a permutation that frequently consecutively samples from the same block mixes a polynomial factor slower than both random scan and most other systematic scans. We now describe this model in detail.

**Variables**    There are $2n$ binary variables grouped into two blocks: $x_1, \ldots, x_n$ and $y_1, \ldots, y_n$. Conditioned on all other variables being false, each variable is equally likely to be true or false. However, the $x$ variables and the $y$ variables contradict each other. As a result, if any of the $x$'s are true, then all of the $y$'s must be false, and if any of the $y$'s are true, then all of the $x$'s must be false. The unnormalized probability distribution for this model is the following.

$$P(x, y) \propto \begin{cases} 0 & \text{if } \exists x_i \text{ true and } \exists y_j \text{ true} \\ 1 & \text{otherwise} \end{cases} \tag{3}$$

This model can be interpreted as a machine learning inference problem in the following way. Each variable represents whether the reasoning in some sentence is sound. The sentences corresponding to $x_1, \ldots, x_n$ and the sentences corresponding to $y_1, \ldots, y_n$ reach contradicting conclusions. If any variable is true, its conclusion is correct, so all of the sentences that reached the opposite conclusion must be not be sound, and their corresponding variables must be false. However, this does not guarantee that all other sentences that reached the same conclusion have sound reasoning, so it is possible for some variables in a block to be true while others are false. Under these assumptions alone, the natural way to model this system is with the two islands distribution in (3).

**State Space**    The states are divided into three groups: states in island $x$ (at least one $x$ variable is true), states in island $y$ (at least one $y$ variable is true), and a single *bridge state* $b$ (all variables are false). The islands are well-connected internally, so the islands mix rapidly. but it is impossible to directly move from one island to the other – the only way to move from one island to the other is through the bridge. To simplify the analysis, we assume that the bridge state has very low mass.[2] This allows us to assume that the chains always move off of the bridge when a variable is sampled.

The bridge is the only way to move from one island to the other, so it acts as a bottleneck. As a result, the efficiency of bridge usage is critical to the mixing time. We will use *bridge efficiency* to refer to the probability that the chain moves to the other island when it reaches the bridge. Because mixing within the islands is rapid in comparison to the time needed to move onto the bridge, the mixing time is inversely proportional to the bridge efficiency of the chain.

**Random Scan**    In random scan, the variable selected after getting on the bridge is independent of the previous variable. As a result, with probability $1/2$, the chain will move onto the other island, and with probability $1/2$, the chain will return to the same island, so the bridge efficiency is $1/2$.

**Best Systematic Scan**    Several different systematic scans achieve the fastest mixing time. One such scan is $x_1, y_1, x_2, y_2, \ldots, x_n, y_n$. Since the sampled variables alternate between the blocks, if the chain moves onto the bridge (necessarily by sampling a variable from the island it was previously on), it will always proceed to sample a variable from the other block, which will cause it to move onto the other island. Thus, the bridge efficiency is $1$. More generally, any systematic scan that alternates between sampling from $x$ variables and sampling from $y$ variables will have a bridge efficiency of $1$.

**Worst Systematic Scan**    Several different systematic scans achieve the slowest mixing time. One such scan is $x_1, \ldots, x_n, y_1, \ldots, y_n$. In this case, if the chain moves onto the bridge, it will almost always proceed to sample a variable from the same block, and return to the same island. In fact, the only way for this chain to move across islands is if it moves from island $x$ to the bridge using transition $x_n$ and then moves to island $y$ using transition $y_1$, or if it moves from island $y$ to the bridge using transition $y_n$ and then moves to island $x$ using transition $x_1$. Thus, only 2 of the $2n$ transitions will cross the bridge, and the bridge efficiency is $1/n$. More generally, any systematic scan that consecutively samples all $x$ variables and then all $y$ variables will have a bridge efficiency of $1/n$.

**Comparison of Mixing Times**    The mixing times of the chains are inversely proportional to the bridge efficiency. As a result, random scan takes twice as long to mix as the best systematic scan, and mixes $n/2$ times faster than the worst systematic scan.

### 3.3 Discrete Pyramid

In the discrete pyramid model (Figure 1c) introduced by Roberts and Rosenthal [20], there are $n$ binary variables $x_i$, and the mass is uniformly distributed over all states where at most one $x_i$ is true. In this model, the mixing time of random scan, $O(n)$, is asymptotically better than that of systematic scan for any permutation, which all have the same mixing time, $O(n^3)$.

## 4 Experiments

In this section, we run several experiments to illustrate the effect of scan order on mixing times. First, in Figure 2a, we plot the mixing times of the models from Section 3 as a function of the number of variables. These experiments validate our results about the asymptotic scaling of the mixing time, as well as show that the scan order can have a significant effect on the mixing time for even small models. (Due to the exponential state space of the two islands model, we modify it slightly to make the computation of mixing times feasible: we simplify the model by only considering the states that are adjacent to the bridge, and assume that the states on each individual island mix instantly.)

In the following experiments, we consider a modified version of the two islands model, in which the mass of the bridge state is set to 0.1 of the mass of the other states to allow the effect of scan order to be clear even for a small number of variables. Figure 2b illustrates the rate at which different scan orders explore this modified model. Due to symmetry, we know that half of the mass should be on each island in the stationary distribution, so getting half of the mass onto the other island is necessary for mixing. This experiment illustrates that random scan and a good systematic scan move to the other island quickly, while a bad systematic scan requires many more iterations.

Figure 2c illustrates the effect that the permutation chosen for systematic scan can have on the mixing time. In this experiment, the mixing time for each permutation was found and plotted in sorted order. For the sequence of dependencies model, there are a small number of good permutations which mix very quickly compared to the other permutations and random scan. However, no permutation is bad compared to random scan. In the two islands model, as we would expect based on the analysis in Section 3, there are a small number of bad permutations which mix very slowly compared to the other permutations and random scan. Some permutations are slightly better than random scan, but none of the scan orders are substantially better. In addition, the mixing times for systematic scan approximately discretized due to the fact that mixing time depends so heavily on the bridge efficiency.

## 5 Relative Bounds on Mixing Times via Conductance

In Section 3, we described two models for which a systematic scan can mix a polynomial factor faster or slower than random scan, thus invalidating conventional wisdom that the scan order does not have an asymptotically significant effect on mixing times. This raises a question of how different the mixing times of different scans can be. In this section, we derive the following weaker – but correct – version of the conjecture stated by Diaconis [3] and Levin et al. [15].

One of the obstacles to proving this result is that the systematic scan chain is not *reversible*. A standard method of handling non-reversible Markov chains is to study a lazy version of the Markov chain instead [15, p. 9]. In the lazy version of a Markov chain, each step has a probability of $1/2$ of staying at the current state, and acts as a normal step otherwise. This is equivalent to stopping at a random time that is distributed as a binomial random variable. Due to the fact that systematic scan is not reversible, our bounds are on the lazy systematic scan, rather than the standard systematic scan.

**Theorem 1.** *For any random scan Gibbs sampler $R$ and lazy systematic scan sampler $S$ with the same stationary distribution $\pi$, their relative mixing times are bounded as follows.*

$$(1/2 - \epsilon)^2 \, t_{mix}(R, \epsilon) \leq 2t_{mix}^2(S, \epsilon) \log\left(\frac{1}{\epsilon \pi_{min}}\right)$$

$$(1/2 - \epsilon)^2 \, t_{mix}(S, \epsilon) \leq \frac{8n^2}{(\min_{x,i} P_i(x,x))^2} t_{mix}^2(R, \epsilon) \log\left(\frac{1}{\epsilon \pi_{min}}\right),$$

*where $P_i$ is the transition matrix corresponding to resampling just variable $i$, and $\pi_{min}$ is the probability of the least likely state in $\pi$.*

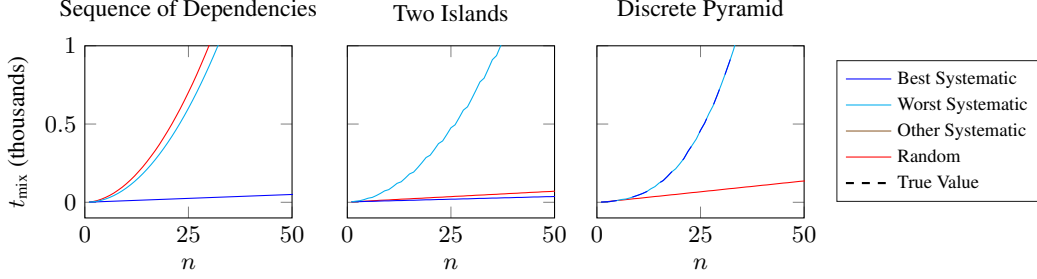

(a) Mixing times for $\epsilon = 1/4$.

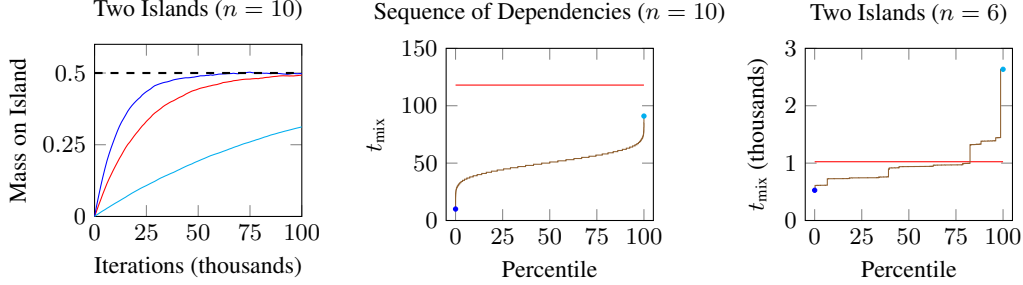

(b) Marginal island mass over time.  (c) Sorted mixing times of different permutations ($\epsilon = 1/4$).

Figure 2: Empirical analysis of the models.

Under mild conditions, namely $\epsilon$ being fixed and the quantities $\log(\pi_{\min}^{-1})$ and $(\min_{x,i} P_i(x,x))^{-1}$ being at most polynomial in $n$, this theorem implies that the choice of scan order can only affect the mixing time by up to polynomial factors in $n$ and $t_{\mathrm{mix}}$. We now outline the proof of this theorem and include full proofs in Appendix D.

In the two islands models, the mixing time of a scan order was determined by its ability to move through a single bridge state that restricted flow. This suggests that a technique with the ability to model the behavior of this bridge state is needed to bound the relative mixing times of different scans. Conductance, also known as the bottleneck ratio, is a topological property of Markov chains used to bound mixing times by considering the flow of mass around the model [15, p. 88]. This ability to model bottlenecks in a Markov chain makes conductance a natural technique both for studying the two islands model and bounding mixing times in general.

More formally, consider a Markov chain on state space $\Omega$ with transition matrix $P$ and stationary distribution $\pi$. The *conductance* of a set $S$ and of the whole chain are respectively defined as

$$\Phi(S) = \frac{\sum_{x \in S, y \notin S} \pi(x) P(x,y)}{\pi(S)} \qquad \Phi_\star = \min_{S : \pi(S) \le \frac{1}{2}} \Phi(S).$$

Conductance can be directly applied to analyze random scan. Let $P_i$ be the transition matrix corresponding to sampling variable $i$. The state space $\Omega$ is used without modification, and the transition matrix is $P = \frac{1}{n} \sum_{i=1}^n P_i$. The stationary distribution is the expected target distribution $\pi$.

On the other hand, conductance cannot be directly applied to systematic scan. Systematic scan is not a homogeneous Markov chain because it uses a sequence of transition matrices rather than a single transition matrix. One standard method of converting systematic scan into a homogeneous Markov chain is to consider each full scan as one step of a Markov chain. However, this makes it difficult to compare with random scan because it completely changes which states are connected by single steps of the transition matrix. To allow systematic and random scan to be compared more easily, we introduce an alternative way of converting systematic scan to a homogeneous Markov chain by augmenting the state space. The augmented state space is $\Psi = \Omega \times [n]$, which represents an ordered pair of the normal state and the index of the variable to be sampled. The transition probability is $P((x,i),(y,j)) = P_i(x,y)s(i,j)$, where $s(i,j) = \mathbb{I}[i+1 \equiv j \pmod n]$ is an indicator that shows if the correct variable will be sampled next.

Additionally, augmenting the state space for random scan allows easier comparison with systematic scan in some cases. For augmented random scan, the state space is also $\Psi = \Omega \times [n]$, the same as for systematic scan. The transition probability is $P\left((x,i),(y,j)\right) = \frac{1}{n} P_i(x,y)$, which means that the next variable to sample is selected uniformly. The stationary distributions of the augmented random scan and systematic scan chains are both $\pi\left((x,i)\right) = n^{-1}\pi(x)$. Because the state space and stationary distribution are the same, augmented random scan and augmented systematic scan can be compared directly, which lets us prove the following lemma.

**Lemma 1.** *For any random scan Gibbs sampler and systematic scan sampler with the same stationary distribution $\pi$, let $\Phi_{RS}$ denote the conductance of the random scan process, let $\Phi_{RS\text{-}A}$ denote the conductance of the augmented random scan process, and let $\Phi_{SS\text{-}A}$ denote the conductance of the augmented systematic scan process. Then,*

$$\frac{1}{2n} \cdot \min_{x,i} P_i(x,x) \cdot \Phi_{RS\text{-}A} \leq \Phi_{SS\text{-}A} \leq \Phi_{RS}.$$

In Lemma 1, the upper bound states that the conductance of systematic scan is no larger than the conductance of random scan. We use this result in the proof of Theorem 1 to show that systematic scan cannot mix too much more quickly than random scan. To prove this upper bound, we show that for any set $S$ under random scan, the set $\hat{S}$ containing the corresponding augmented states for systematic scan will have the same conductance under systematic scan as $S$ had under random scan.

The lower bound in Lemma 1 states that the conductance of systematic scan is no smaller than a function of the conductance of augmented random scan. This function depends on the number of variables $n$ and $\min_{x,i} P_i(x,x)$, which is the minimum holding probability of any state. To prove this lower bound, we show that for any set $S$ under augmented systematic scan, we can bound its conductance under augmented random scan.

There are well-known bounds on the mixing time of a Markov chain in terms of its conductance, which we state in Theorem 2 [15, pp. 89, 235].

**Theorem 2.** *For any lazy or reversible Markov chain,*

$$\frac{1/2 - \epsilon}{\Phi_\star} \leq t_{mix}(\epsilon) \leq \frac{2}{\Phi_\star^2} \log\left(\frac{1}{\epsilon \pi_{min}}\right).$$

It is straightforward to prove the result of Theorem 1 by combining the bounds from Theorem 2 with the conductance bounds from Lemma 1.

# 6 Conclusion

We studied the effect of scan order on mixing times of Gibbs samplers, and found that for particular models, the scan order can have an asymptotic effect on the mixing times. These models invalidate conventional wisdom about scan order and show that we cannot freely change scan orders without considering the resulting changes in mixing times. In addition, we found bounds on the mixing times of different scan orders, which replaces a common conjecture about the mixing times of random scan and systematic scan.

**Acknowledgments**

The authors acknowledge the support of: DARPA FA8750-12-2-0335; NSF IIS-1247701; NSF CCF-1111943; DOE 108845; NSF CCF-1337375; DARPA FA8750-13-2-0039; NSF IIS-1353606; ONR N000141210041 and N000141310129; NIH U54EB020405; NSF DGE-114747; DARPA's SIMPLEX program; Oracle; NVIDIA; Huawei; SAP Labs; Sloan Research Fellowship; Moore Foundation; American Family Insurance; Google; and Toshiba. The views and conclusions expressed in this material are those of the authors and should not be interpreted as necessarily representing the official policies or endorsements, either expressed or implied, of DARPA, AFRL, NSF, ONR, NIH, or the U.S. Government.

## Footnotes

[1]We discuss the necessary magnitude of $M$ in Appendix B

[2]We show that the same asymptotic result holds without this assumption in Appendix C.

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
