[Supplementary Material]

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

_{\text{mix}}(R)$ | $\min\limits_{\alpha} t_{\text{mix}}(S_\alpha)$ | $\max\limits_{\alpha} t_{\text{mix}}(S_\alpha)$ |
|---|---|---|---|
| Sequence of Dependencies | $n^2$ | $n$ | $n^2$ |
| Two Islands | $2^n$ | $2^n$ | $n2^n$ |
| Discrete Pyramid [20] | $n$ | $n^3$ | $n^3$ |
| Memorize and Repeat | $n^3$ | $n^2$ | $n^2$ |
| Soft Dependencies | $n^{3/2}$ | $n$ | $n^2$ |

The *mixing time* is the minimum number of steps needed to guarantee that the total variation distance between the true and estimated distributions is below a given threshold $\epsilon$ from any starting distribution. Formally, the *mixing time* of a stochastic process $P$ with transition matrix $P^{(t)}$ after $t$ steps and stationary distribution $\pi$ is [15, p. 55]

$$t_{\text{mix}}(P, \epsilon) = \min \left\{ t : \max_{\mu} \|P^{(t)}\mu - \pi\|_{\text{TV}} \leq \epsilon \right\},$$

where the maximum is taken over the distribution $\mu$ of the initial state of the process. When comparing the statistical efficiency of systematic scan and random scan, it would be useful to establish, for any systematic scan process $S$ and random scan process $R$ on the same $n$-variable model, a relative bound of the form

$$F_1(\epsilon, n, t_{\text{mix}}(R, \epsilon)) \leq t_{\text{mix}}(S, \epsilon) \leq F_2(\epsilon, n, t_{\text{

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

# A   Additional Models in Which Scan Order Matters

Table 2: Models Classified by Relative Mixing Times

$\min_\alpha t_{\mathrm{mix}}(S_\alpha)$ versus $t_{\mathrm{mix}}(R)$

| | | $\ll$ | $\approx$ | $\gg$ |
|---|---|---|---|---|
| $\max_\alpha t_{\mathrm{mix}}(S_\alpha)$ | $\ll$ | Memorize and Repeat | $\times$ | $\times$ |
| versus | $\approx$ | Sequence of Dependencies | Known Cases | $\times$ |
| $t_{\mathrm{mix}}(R)$ | $\gg$ | Soft Dependencies | Two Islands | Discrete Pyramid [20] |

Table 2 classifies the models based on the mixing times of the best systematic scan and worst systematic scan relative to random scan. For models in the first column, the best systematic scan mixes asymptotically faster than random scan. For models in the second column, they differ only up to logarithmic factors. For models in the third column, the best systematic scan mixes asymptotically slower than random scan. The rows are classified in the same way based on the worst systematic scan instead of the best systematic scan.

The models in Section 3 display possible problems that can occur when changing scan orders. However, two behaviors were not shown by the models in Section 3: a case where random scan is asymptotically worse than all systematic scans, and a case where random scan is asymptotically better than a systematic scan and asymptotically worse than another systematic scan. This section gives two additional models that exhibit these behaviors, along with a more detailed explanation of the discrete pyramid model. These two models are more complicated than the models in Section 3 due to the fact that they require the number of states for each variable to grow with $n$. Combined with the models in Section 3, these models show that all logically consistent asymptotic behaviors (that is, behaviors where the worst systematic scan is no better than the best systematic scan) are possible.

## A.1   Discrete Pyramid

The discrete pyramid model (Figure 1c) was first described by Roberts and Rosenthal [20] and is included for completeness to show that it is possible for random scan to mix asymptotically faster than all systematic scans.

**Variables**   There are $n$ binary variables $x_1, \ldots, x_n$. Conditioned on all other variables being false, each variable is equally likely to be true or false. However, the variables are all contradictory, and at most one variable can be true. This model can also be interpreted as an $n$ islands model, where there are $n$ well-connected regions (which consist of a single state) connected by a single bridge state.

**State Space**   There are $n+1$ states $s_0, s_1, \ldots, s_n$ with nonzero probability. $s_0$ is the state where all variables are false, and for all other $i$, $s_i$ is the state where variable $x_i$ is true and all other variables are false.

**Random Scan**   The worst-case total variation occurs when the starting state is not $x_0$. Suppose that the starting state is $s_k$. $x_k$ will be selected with probability $1/n$, and in this case, the state will change to $x_0$ with probability $1/2$. Thus, the number of samples needed to leave $x_k$ is distributed as a Geometric random variable with a mean of $2n$. This means that within $O(n)$ steps, the chain will have left the initial state with high probability. Once the chain has reached $x_0$, each step has a probability of $1/2$ of leaving $x_0$ and uniformly going to another state. Thus, $O(1)$ steps are sufficient for the chain to mix after it reaches $x_0$. In total, the chain mixes in $O(n)$ updates.

**All Systematic Scans**   Once again, the worst-case total variation occurs when the starting state is not $x_0$. Suppose that the starting state is $s_k$. A systematic scan step will change nothing until $x_k$ is reached. Then, with probability $1/2$, the state will remain $s_k$, and with probability $1/2$ the state will change to $x_0$. If it does move to $x_0$, the scan will continue, and move away from $x_0$ with probability $1/2$ at each step. Thus, each time the scan reaches the current state, the state will advance $Z$ steps, where $Z$ is a Geometric random variable with rate $1/2$. This corresponds to a weighted random walk on a circle, which is known to have a mixing time of $O(n^2)$. Notice that each step of the random walk actually requires one full sweep, so $O(n^3)$ steps are needed for any systematic scan to mix.

Figure 3: Memorize and Repeat Model

## A.2 Memorize and Repeat

We introduce the memorize and repeat model to show that it is possible for all systematic scans to be asymptotically better than random scan.

**Description** In this example (Figure 3), there are two types of states: states memorizing a permutation, and states requiring the memorized permutation to be repeated. In the stationary distribution, almost all of the mass is on the state where a permutation has been memorized and repeated, and the probability of the states increase exponentially from left to right. There are $n$ variables $x_1, \ldots, x_n$. In the memorize phase, sampling a variable will allow the state to change as long as the variable has not been sampled before. In the repeat phase, a sampling a variable will only allow the state to change if it is the next variable in the memorized permutation. The repeat phase is similar to the sequence of dependencies example, with the variables rearranged to match the memorized permutation.

In the worst-case starting distribution, the wrong permutation can already be memorized, which results in no asymptotic gap between mixing times. To allow the difference in mixing time to appear even in the worst-case analysis, we repeat the memorize and repeat process $n$ times.

**Random Scan** Random scan encounters the coupon collector problem to get through the memorize phase, so $O(n \log n)$ steps are needed. Afterwards, the repeat phase is equivalent to a sequence of dependencies, which requires $O(n^2)$ steps. Thus, to get through one memorize and repeat chain, $O(n^2)$ steps are needed. This mechanism is repeated $n$ times, so $O(n^3)$ steps are needed.

**All Systematic Scans** If a systematic scan starts at the beginning of a memorize sequence, $2n$ steps are sufficient to move through the memorize and repeat chain ($n$ steps to memorize, $n$ steps to repeat). However, when analyzing the worst-case total variation, we must consider the case where the wrong permutation has already been memorized, which requires $O(n^2)$ steps to move through. However, this can only happen once, so in the worst case $O(n^2)$ steps are needed to move through the one incorrect permutation, and $O(n)$ steps are needed for the remaining $n - 1$ memorize and repeat chains. Thus, $O(n^2)$ steps are needed in total.

**State Space** To simplify the explanation of the state space and formulation as a Gibbs sampler, this explanation is for only one memorize and repeat cycle. In the memorize phase, the states are labeled by the part of the permutation that has already been memorized. In the repeat phase, the states are labeled by the remainder of the permutation that still has to be repeated.

**Formulation as a Gibbs Sampler** There are $n$ integer-valued variables $x_1, \ldots, x_n$ with values ranging from 0 to $n + 1$. In the initial state that does not have anything memorized, all variables have a value of 0. In the memorize phase, any variable that has not yet been memorized has a value of 0, and any variable that has been memorized stores its index within the permutation. This means that for states in the memorize phase with $i$ variables memorized, exactly one variable will have a value of $j$, for each integer $j$ from 1 to $i$, and the remaining $n - i$ variables that have not been memorized have a value of 0. As more variables are memorized, the states become exponentially more likely. Next, in the repeat phase, when a variable is used, its value changes to $n + 1$. This means that for states in the repeat phase with $i$ variables repeated, the $i$ variables that have been repeated have a value of

$n + 1$, and the remaining $n - i$ variables that still have to be repeated have a unique integer from $i + 1$ to $n$. As more variables are repeated, the states become exponentially more likely. The log probability distribution is the following, where $x$ denotes $(x_1, \ldots, x_n)$, $M$ is a very large constant, $Z$ is the normalizing constant, and $\sigma(a_1, \ldots, a_n)$ denotes the set of all permutations of $(a_1, \cdots, a_n)$.

$$\log_M P(x) = -\log_M Z + \begin{cases} \max(x) & \text{if } x \in \sigma(0, \ldots, 0, \ 1, \ldots, i) \\ n + \sum_{i=1}^{n} \mathbb{I}[x_i = n+1] & \text{if } x \in \sigma(i+1, \ldots, n, \ n+1, \ldots, n+1) \\ -\infty & \text{otherwise} \end{cases}$$

$$= -\log_M Z + \begin{cases} \text{(Number of Memorized Variables)} & \text{if valid memorize state} \\ n + \text{(Number of Repeated Variables)} & \text{if valid repeat state} \\ -\infty & \text{otherwise} \end{cases}$$

First, for states in the memorize phase, if a variable that has not yet been memorized is sampled, then its value will almost always change to $i + 1$, where $i$ is the number of variables already memorized, due to the large value of $M$. However, sampling a variable that has already been memorized will not change its value because all other values will either result in an invalid state (and have a probability of 0) or be significantly less likely due to the large value of $M$. Next, for states in the repeat phase, if the correct variable is sampled, then its value will almost always change to $n + 1$. However, sampling the wrong variable will not change its value because all other values result in an invalid state or decrease the probability of the state.

## A.3 Soft Dependencies

In the soft dependencies model, some systematic scans mix asymptotically faster than random scan, and some systematic scans mix asymptotically slower.

**Description**   This example resembles the memorize portion of the previous example. However, no repeat phase is included, and only some permutations are accepted. In particular, a permutation is accepted only if each element of the permutation is followed by an element that is in the next $\sqrt{n}$ unused variables that come after it (mod $n$).

More formally, consider a permutation $(a_1, \ldots, a_n)$ of $(1, \ldots, n)$. The permutation is accepted only if the following holds for all $i \in \{1, \ldots, n-1\}$.

$$\sqrt{n} \geq |\{\, j \in \{1, \ldots, n\} \mid j > i + 1 \text{ and } a_j - a_i \ (\text{mod } n) < a_{i+1} - a_i \ (\text{mod } n) \,\}|$$

In this condition, the requirement that $j > i + 1$ means that $a_j$ is an unused element, and the requirement that $a_j - a_i \ (\text{mod } n) < a_{i+1} - a_i \ (\text{mod } n)$ means that starting from $a_i$, $a_j$ is reached before $a_{i+1}$. These two requirements imply that $a_{i+1}$ is within the next $\sqrt{n}$ unused variables following $a_i$. This condition is equivalent to the following.

$$\sqrt{n} \geq \begin{cases} \sum_{j=i+2}^{n} \mathbb{I}[a_i < a_j < a_{i+1}] & \text{if } a_i < a_{i+1} \\ \sum_{j=i+2}^{n} \mathbb{I}[a_i < a_j \text{ or } a_j < a_{i+1}] & \text{if } a_i > a_{i+1} \end{cases}$$

**Random Scan**   First, consider when there are at least $\sqrt{n}$ remaining transitions. The first $n - \sqrt{n} = O(n)$ transitions will have this property. During this period, random scan advances to the next state with probability $n/\sqrt{n} = \sqrt{n}$. Thus, $O(n\sqrt{n})$ steps are needed to get until there are fewer than $\sqrt{n}$ remaining transitions. Once there are fewer than $\sqrt{n}$ remaining transitions, random scan needs $O(n)$ steps to make each transition, so $O(n\sqrt{n})$ more steps are needed. In total, $O(n\sqrt{n})$ steps are needed.

**Best Systematic Scan**   The best systematic scan uses the order $x_1, \ldots, x_n$. This uses $n$ steps to give a valid permutation, and thus mixes in $n$ steps.

**Worst Systematic Scan**   The worst systematic scan uses the order $x_n, x_{n-1}, \ldots, x_1$. After a transition is taken, nearly a full scan is needed until the next valid transition will be reached, so $O(n^2)$ steps are needed.

**State Space**   The state space has a similar form to the memorize portion of the previous example. The states are labeled by the portion of the permutation that has already been given, but only prefixes to valid permutations are accepted.

**Formulation as a Gibbs Sampler**   There are $n$ integer-valued variables $x_1, \ldots, x_n$ with values ranging from 0 to $n$. In the initial state that does not have anything memorized, all variables have a value of 0. Then, as the permutation is memorized, the variables that have not been used yet have a value of 0, and any variable that has been used stores its index within the permutation.

In addition, to merge the permutations together into one state, all variables need to be sampled one more time (note that this does not require the variables to be sampled in a particular order). This extra sampling process only adds lower order terms and constant factors to the mixing time. Once each variable is sampled again, its value is changed to $n$.

The log probability distribution is the following, where $x$ denotes $(x_1, \ldots, x_n)$, $M$ is a very large constant, $Z$ is the normalizing constant, and $\sigma(a_1, \ldots, a_n)$ denotes the set of all permutations of $(a_1, \ldots, a_n)$.

$$\log_M P(x) = -\log_M Z + \begin{cases} \max(x) & \text{if } x \in \sigma(0, \ldots, 0,\ 1, \ldots, i) \text{ and is valid} \\ n - 1 + \sum_{i=1}^n \mathbb{I}[x_i = n] & \text{if at least two of } (x_1, \ldots, x_n) \text{ have a value of } n \\ -\infty & \text{otherwise} \end{cases}$$

$$= -\log_M Z + \begin{cases} \text{(Number of Memorized Variables)} & \text{if valid memorize state} \\ n - 1 + \text{(Number of Merged Variables)} & \text{if valid merge state} \\ -\infty & \text{otherwise} \end{cases}$$

First, in the memorize phase, sampling a valid unused variable will almost always change its value to $i + 1$, where $i$ is the number of variables already memorized, due to the large value of $M$. However, sampling a variable that has already been memorized or sampling a variable that least to an invalid permutation will not change its value. Then, when the last variable in the permutation is sampled, its value changes to $n$. Afterwards, sampling any variable will also change its value to $n$.

## B   Priors in the Sequence of Dependencies

To simplify the analysis of the sequence of dependencies, we assumed that the priors on the variables were strong enough such sampling the correct variable essentially always caused the state to advance. We now analyze how large $M$ (the strength of the priors) needs to be in order to allow the best systematic scan to mix in $O(n)$ time. In this section, we show that if $M = \Omega(n)$, then the best systematic scan mixes in $O(n)$ time, and if $M = o(n)$, then the best systematic scan cannot mix $O(n^2)$ time.

First, suppose that $M = \Omega(n)$. This means that $M \geq cn$ for some $c > 0$ and sufficiently large $n$. The probability of transitioning from $s_{i-1}$ to $s_i$ when variable $x_i$ is sampled is

$$\frac{M}{1 + M} \geq \frac{cn}{1 + cn} = 1 - \frac{1}{1 + cn} > 1 - \frac{1}{cn}.$$

The probability of transitioning from $s_0$ to $s_n$ after sampling the sequence $x_1, \ldots, x_n$ is at least $(1 - 1/cn)^n$, which limits to $e^{-1/c}$ for large $n$. In other words, if $M = \Omega(n)$, then a single sweep of the best systematic scan will reach $s_n$ with a probability that does not approach 0. Thus, a constant number of sweeps is sufficient to reach $s_n$ with high probability, which is equivalent to mixing.

On the other hand, suppose that $M = o(n)$. Then, for any $c > 0$, $M < cn$ for sufficiently large $n$. This means that as $n$ increases, the probability that a single sweep of the best systematic scan will reach $s_n$ will go to 0, so no constant number of sweeps will be sufficient to mix.

## C   Bridge Efficiency with Normal Mass on Bridge

In the analysis of the two islands model, we assumed that the bridge has negligible mass. We now analyze the mixing times without the assumption that the bridge has small mass. The same asymptotic behavior still results.

Even when the bridge has the same mass as the other states, it still acts as a bottleneck to the model. For sufficiently large $n$, the islands will mix rapidly in comparison reaching the bridge, so the mixing time is still inversely proportional to the bridge efficiency. However, when the bridge has the same mass as the other states, sampling a variable while on the bridge will only have a $1/2$ chance of moving off of the bridge.

**Random Scan** In random scan, the variables that are sampled are completely independent, so the bridge efficiency is still $1/2$.

**Best Systematic Scan** Consider the scan $x_1, y_1, x_2, y_2, \ldots$. Suppose the sampling $x_1$ changes the state to the bridge state – a similar analysis will apply for any other variable. The next variable is $y_1$, which will change the state to the other island with probability $1/2$. Afterwards, $x_2$ is sampled, which will change the state to the same island with probability $1/4$. Then, $y_2$ is sampled, which will change the state to the other island with probability $1/8$. Thus, the probability of moving to the other island is $1/2 + 1/8 + 1/32 + \ldots = 2/3$.

**Worst Systematic Scan** Consider the scan $x_1, \ldots, x_n, y_1, \ldots, y_n$. Suppose that sampling $x_n$ changes the state to the bridge state. For large $n$, the probability of leaving the bridge state onto island $y$ approaches 1. Next, suppose that sampling $x_{n-1}$ changes the state to the bridge state. There is a probability of $1/2$ of moving back to island $x$ via $x_n$, but the chain will move onto island $y$ otherwise. In general, moving onto the bridge via $x_i$ or $y_i$ will result in moving to the island with probability $2^{-n+i}$. The average probability of moving onto the other island is then $2/n$.

## D    Proofs for Section 5

In this section, we prove our relative bounds on mixing times (Theorem 1), along with related claims and lemmas.

**Claim 1.** *The stationary distribution of augmented random and systematic scan is*

$$\pi\left((x, i)\right) = \frac{1}{n}\pi(x)$$

*Proof.* We prove this claim by showing that applying the transition matrix for augmented random scan or augmented systematic scan does not change this distribution.

For augmented random scan,

$$\sum_{x,i} \pi((x,i))P((x,i),(y,j)) = \sum_{x,i} \frac{\pi(x)}{n} \cdot P((x,i),(y,j)) = \frac{1}{n}\sum_{x,i} \pi(x) \cdot \frac{1}{n}P_i(x,y)$$

$$= \frac{1}{n}\sum_{i=1}^{n}\left[\sum_{x\in\Omega}(\pi(x)P_i(x,y)) \cdot \frac{1}{n}\right]$$

$$= \frac{1}{n}\sum_{i=1}^{n}\left[\pi(y) \cdot \frac{1}{n}\right] = \frac{1}{n}\pi(y)$$

$$= \pi((y,j))$$

For augmented systematic scan,

$$\sum_{x,i} \pi((x,i))P((x,i),(y,j)) = \sum_{x,i} \frac{\pi(x)}{n} \cdot P((x,i),(y,j)) = \frac{1}{n}\sum_{x,i} \pi(x) \cdot P_i(x,y)s(i,j)$$

$$= \frac{1}{n}\sum_{i=1}^{n}\left[\sum_{x\in\Omega}(\pi(x)P_i(x,y))\, s(i,j)\right]$$

$$= \frac{1}{n}\sum_{i=1}^{n}\left[\pi(y)s(i,j)\right] = \frac{1}{n}\pi(y)$$

$$= \pi((y,j))$$

$\square$

**Lemma 1.** *For any random scan Gibbs sampler and systematic scan sampler with the same stationary distribution $\pi$, let $\Phi_{RS}$ denote the conductance of the random scan process, let $\Phi_{RS\text{-}A}$ denote the conductance of the augmented random scan process, and let $\Phi_{SS\text{-}A}$ denote the conductance of the augmented systematic scan process. Then,*

$$\frac{1}{2n} \cdot \min_{x,i} P_i(x,x) \cdot \Phi_{RS\text{-}A} \leq \Phi_{SS\text{-}A} \leq \Phi_{RS}.$$

*Proof.*

**Upper Bound:** The conductance of the whole chain is the smallest conductance of any set with mass no larger than $\frac{1}{2}$. Then, to prove that this inequality holds, we will show that for any set $S \in \Omega$ with mass no larger than $\frac{1}{2}$, there exists a set $T \in \Psi$ with mass no larger than $\frac{1}{2}$ such that the conductance of $S$ under random scan is the same as the conductance of $T$ under augmented systematic scan.

From the standpoint of random scan, consider a set $S \in \Omega$ with mass no larger than $\frac{1}{2}$. The conductance is

$$\Phi_{\text{RS}}(S) = \frac{\sum_{x \in S} \sum_{y \in S^c} \pi(x) P(x,y)}{\pi(S)}$$
$$= \frac{\frac{1}{n} \sum_{i=1}^{n} \sum_{x \in S} \sum_{y \in S^c} \pi(x) P_i(x,y)}{\pi(S)}$$

Then, for augmented systematic scan, consider the set $T = \{(x,i) : x \in S, i \in [n]\}$. First, notice that $\pi(T) = \pi(S) \leq \frac{1}{2}$. The conductance is

$$\Phi_{\text{SS-A}}(T) = \frac{\sum_{(x,i) \in T} \sum_{(y,j) \in T^c} \pi((x,i)) P((x,i),(y,j))}{\pi(T)}$$
$$= \frac{\sum_{i=1}^{n} \sum_{j=1}^{n} \sum_{x \in S} \sum_{y \in S^c} \pi((x,i)) P((x,i),(y,j))}{\pi(S)}$$
$$= \frac{1}{n} \frac{\sum_{i=1}^{n} \sum_{j=1}^{n} \sum_{x \in S} \sum_{y \in S^c} \pi(x) P_i(x,y) s(i,j)}{\pi(S)}$$
$$= \frac{1}{n} \frac{\sum_{i=1}^{n} \sum_{x \in S} \sum_{y \in S^c} \pi(x) P_i(x,y)}{\pi(S)}$$
$$= \Phi_{\text{RS}}(S)$$

This implies that for any $S \in \Omega$ with $\pi(S) \leq \frac{1}{2}$, there exists $T \in \Psi$ with $\pi(T) \leq \frac{1}{2}$ such that $\Phi_{\text{RS}}(S) = \Phi_{\text{SS}}(T)$. Therefore, $\Phi_{\text{SS-A}} \leq \Phi_{\text{RS}}$.

**Lower Bound:** In this proof, we will work with the *flow* between two sets

$$Q(A,B) = \sum_{x \in A, y \in B} \pi(x) P(x,y).$$

Notice that the conductance of a set can then be defined as

$$\Phi(S) = \frac{Q(S, S^c)}{\pi(S)}.$$

To prove that this inequality holds for the whole chain, we will show that the same inequality holds for any set $S \in \Psi$. Consider some arbitrary state $x \in \Omega$. Flow can leave from the corresponding augmented states in two ways: flowing from some $(x,i) \in S$ to $(x,j) \in S^c$ or flowing from $(x,i) \in S$ to $(y,j) \in S^c$, where $y \neq x$ ($x$ and $y$ differ in only variable $i$). Let $S_x = \{(x,i) \in S\}$, and let $S_x^c = \{(x,i) \in S^c\}$. These two components can be written as $Q(S_x, S_x^c)$ and $\sum_{y \neq x} Q(S_x, S_y^c)$.

Now, we will find upper bounds for the random scan flows and lower bounds for the systematic scan flows. In the following statements, it is implicit that $y \neq x$, and $\gamma = \min_{x,i} P_i(x,x)$ will denote the minimum holding probability.

First, we bound the amount of flow from $(x, i) \in S$ to $(x, j) \in S^c$. For augmented random scan, the following upper bound holds.

$$Q_{\text{RS}}(S_x, S_x^c) = \sum_{(x,i) \in S_x, (x,j) \in S_x^c} \pi((x,i)) P((x,i),(x,j))$$

$$= \sum_{(x,i) \in S_x, (x,j) \in S_x^c} \frac{1}{n} \pi(x) \cdot \frac{1}{n} P_i(x,x)$$

$$\leq \sum_{(x,i) \in S_x, (x,j) \in S_x^c} \frac{1}{n} \pi(x) \cdot \frac{1}{n}$$

$$= \frac{|S_x|}{n} \pi(x) \frac{n - |S_x|}{n}$$

$$\leq \begin{cases} \frac{1}{4}\pi(x) & \text{if } |S_x| \neq 0, n \\ 0 & \text{if } |S_x| = 0, n \end{cases}$$

For augmented systematic scan, the following lower bound holds.

$$Q_{\text{SS}}(S_x, S_x^c) = \sum_{(x,i) \in S_x, (x,j) \in S_x^c} \pi((x,i)) P((x,i),(x,j))$$

$$= \sum_{(x,i) \in S_x, (x,j) \in S_x^c} \frac{1}{n} \pi(x) \cdot P_i(x,x) s(i,j)$$

$$\geq \sum_{(x,i) \in S_x, (x,j) \in S_x^c} \frac{1}{n} \pi(x) \cdot \gamma \cdot s(i,j)$$

$$\geq \begin{cases} \frac{1}{n}\pi(x)\gamma & \text{if } |S_x| \neq 0, n \\ 0 & \text{if } |S_x| = 0, n \end{cases}$$

Similarly,

$$Q_{\text{SS}}(S_y, S_y^c) \geq \begin{cases} \frac{1}{n}\pi(y)\gamma & \text{if } |S_y| \neq 0, n \\ 0 & \text{if } |S_y| = 0, n \end{cases}$$

Now, we bound the amount of flow from $x$ to $y$ for $y \neq x$. Note that $P_i(x, y) = 0$ for all $i$ if $x$ and $y$ differ in more than one variable. As a result, we will assume that $x$ and $y$ differ in only variable $i$ for the next two bounds. For augmented random scan, the following upper bound holds.

$$Q_{\text{RS}}(S_x, S_y^c) = \begin{cases} \frac{1}{n}\pi(x) P_i(x,y) \frac{n - |S_y|}{n} & \text{if } (x,i) \in S \\ 0 & \text{if } (x,i) \notin S \end{cases}$$

$$\leq \begin{cases} \frac{1}{n}\pi(x) P_i(x,y) & \text{if } (x,i) \in S \text{ and } |S_y^c| \neq 0 \\ 0 & \text{if } (x,i) \notin S \text{ or } |S_y^c| = 0 \end{cases}$$

In the derivation of the next bound, note that if $|S_y^c| = n$, then we are guaranteed that $(y, i + 1 (\text{mod } n)) \in S_y^c$. For augmented systematic scan, the following lower bound holds.

$$Q_{\text{SS}}(S_x, S_y^c) = \begin{cases} \frac{1}{n}\pi(x) P_i(x,y) & \text{if } (x,i) \in S \text{ and } (y, i+1) \in S^c \\ 0 & \text{otherwise} \end{cases}$$

$$\geq \begin{cases} \frac{1}{n}\pi(x) P_i(x,y) & \text{if } (x,i) \in S \text{ and } |S_y^c| = n \\ 0 & \text{otherwise} \end{cases}$$

Now, we can derive several inequalities between the augmented random scan flow and the augmented systematic scan flow as direct consequences of the bounds we just found. First, we bound the relative flows from $S_x$ to $S_x^c$.

$$Q_{\text{SS}}(S_x, S_x^c) \geq \frac{4\gamma}{n} Q_{\text{RS}}(S_x, S_x^c) \geq \frac{\gamma}{n} Q_{\text{RS}}(S_x, S_x^c)$$

Next, we bound the relative flows from $x$ to $y$, where $x$ and $y$ differ in exactly variable $i$.

$$Q_{\text{SS}}(S_x, S_y^c) + \frac{1}{n} P_i(y,x) Q_{\text{SS}}(S_y, S_y^c)$$

$$\geq \begin{cases} \frac{1}{n}\pi(x)P_i(x,y) & \text{if } (x,i) \in S \text{ and } |S_y^c| = n \\ \frac{1}{n}P_i(y,x) \cdot \frac{1}{n}\pi(y)\gamma & \text{if } (x,i) \in S \text{ and } |S_y^c| \neq 0, n \\ 0 & \text{otherwise} \end{cases}$$

$$= \begin{cases} \frac{1}{n}\pi(x)P_i(x,y) & \text{if } (x,i) \in S \text{ and } |S_y^c| = n \\ \frac{1}{n^2}\pi(x)P_i(x,y)\gamma & \text{if } (x,i) \in S \text{ and } |S_y^c| \neq 0, n \\ 0 & \text{otherwise} \end{cases}$$

$$\geq \begin{cases} \frac{1}{n^2}\pi(x)P_i(x,y)\gamma & \text{if } (x,i) \in S \text{ and } |S_y^c| \neq 0 \\ 0 & \text{otherwise} \end{cases}$$

$$= \frac{\gamma}{n} \begin{cases} \frac{1}{n}\pi(x)P_i(x,y) & \text{if } (x,i) \in S \text{ and } |S_y^c| \neq 0 \\ 0 & \text{otherwise} \end{cases}$$

$$\geq \frac{\gamma}{n} \Phi_{\text{RS}}(S_x, S_y^c)$$

Finally, we bound the relative flows.

$$Q_{\text{SS}}(S, S^c) = \sum_{x \in \Omega}\sum_{y \in \Omega} Q_{\text{SS}}(S_x, S_y^c) = \sum_{x \in \Omega}\left( Q_{\text{SS}}(S_x, S_x^c) + \sum_{y \neq x} Q_{\text{SS}}(S_x, S_y^c) \right)$$

$$\geq \frac{1}{2}\sum_{x \in \Omega}\left( 2Q_{\text{SS}}(S_x, S_x^c) + \sum_{y \neq x} Q_{\text{SS}}(S_x, S_y^c) \right)$$

$$\frac{1}{2}\sum_{x \in \Omega}\left( Q_{\text{SS}}(S_x, S_x^c) + \sum_{y \in \Omega}\sum_{i=1}^{n}\frac{1}{n}P_i(y,x)Q_{\text{SS}}(S_y, S_y^c) + \sum_{y \neq x} Q_{\text{SS}}(S_x, S_y^c) \right)$$

$$\geq \frac{1}{2}\sum_{x \in \Omega}\left( Q_{\text{SS}}(S_x, S_x^c) + \sum_{y \neq x}\left( Q_{\text{SS}}(S_x, S_y^c) + \sum_{i=1}^{n}\frac{1}{n}P_i(y,x)Q_{\text{SS}}(S_y, S_y^c) \right) \right)$$

$$\geq \frac{1}{2}\sum_{x \in \Omega}\left( \frac{\gamma}{n}Q_{\text{RS}}(S_x, S_x^c) + \frac{\gamma}{n}\sum_{y \neq x} Q_{\text{RS}}(S_x, S_y^c) \right)$$

$$\geq \frac{\gamma}{2n}\sum_{x \in \Omega}\left( Q_{\text{RS}}(S_x, S_x^c) + \sum_{y \neq x} Q_{\text{RS}}(S_x, S_y^c) \right)$$

$$= \frac{\gamma}{2n}Q_{\text{SS}}(S, S^c)$$

The mass of $S$ is the same for augmented random scan and augmented systematic scan, so the same inequality holds for the conductances of the sets. Finally, because this inequality holds for any set $S$, the inequality also holds for the conductance of the whole chain. $\qquad\square$

**Theorem 2.** *For any lazy or reversible Markov chain,*

$$\frac{1/2 - \epsilon}{\Phi_\star} \leq t_{mix}(\epsilon) \leq \frac{2}{\Phi_\star^2}\log\left( \frac{1}{\epsilon\pi_{min}} \right).$$

*Proof.* The lower bound of this inequality,

$$\frac{1/2 - \epsilon}{\Phi_\star} \leq t_{\text{mix}}(\epsilon)$$

is is shown by Theorem 7.3 of [15] and holds for any Markov chain – that is, it does not actually require the Markov chain to be lazy or reversible.

The upper bound of this inequality,

$$t_{\mathrm{mix}}(\epsilon) \leq \frac{2}{\Phi_\star^2} \log\left(\frac{1}{\epsilon\pi_{\min}}\right),$$

is shown by Theorem 17.10 of [15]. □

**Theorem 1.** *For any random scan Gibbs sampler $R$ and lazy systematic scan sampler $S$ with the same stationary distribution $\pi$, their relative mixing times are bounded as follows.*

$$(1/2 - \epsilon)^2 \, t_{mix}(R, \epsilon) \leq 2t_{mix}^2(S, \epsilon) \log\left(\frac{1}{\epsilon\pi_{min}}\right)$$

$$(1/2 - \epsilon)^2 \, t_{mix}(S, \epsilon) \leq \frac{8n^2}{(\min_{x,i} P_i(x,x))^2} t_{mix}^2(R, \epsilon) \log\left(\frac{1}{\epsilon\pi_{min}}\right),$$

*where $P_i$ is the transition matrix corresponding to resampling just variable $i$, and $\pi_{min}$ is the probability of the least likely state in $\pi$.*

*Proof.*

**Upper Bound for Random Scan:** First, we upper bound the mixing time of random scan.

$$t_{\mathrm{mix}}(R, \epsilon) \leq \frac{2}{\Phi_{\mathrm{RS}}^2} \log\left(\frac{1}{\epsilon\pi_{\min}}\right)$$

Next, we lower bound the mixing time for systematic scan.

$$t_{\mathrm{mix}}(S, \epsilon) \geq \frac{1/2 - \epsilon}{\Phi_{\mathrm{SS\text{-}A}}} \geq \frac{1/2 - \epsilon}{\Phi_{\mathrm{RS}}}$$

This theorem results from algebraic manipulation of the previous two inequalities.

$$t_{\mathrm{mix}}^2(S, \epsilon) \geq \frac{(1/2 - \epsilon)^2}{\Phi_{\mathrm{RS}}^2}$$

$$(1/2 - \epsilon)^2 \, t_{\mathrm{mix}}(R, \epsilon) \leq 2t_{\mathrm{mix}}^2(S, \epsilon) \log\left(\frac{1}{\epsilon\pi_{\min}}\right)$$

**Upper Bound for Systematic Scan:** First, we lower bound the mixing time for random scan.

$$t_{\mathrm{mix}}(R, \epsilon) \geq \frac{1/2 - \epsilon}{\Phi_{\mathrm{RS\text{-}A}}}$$

$$t_{\mathrm{mix}}(R, \epsilon)^2 \geq \frac{(1/2 - \epsilon)^2}{\Phi_{\mathrm{RS\text{-}A}}^2}$$

Next, we manipulate the lower bound of Lemma 1.

$$\Phi_{\mathrm{SS}} \geq \frac{1}{2n} \cdot \min_{x,i} P_i(x,x) \cdot \Phi_{\mathrm{RS\text{-}A}}$$

$$\frac{1}{\Phi_{\mathrm{SS}}^2} \leq \frac{4n^2}{(\min_{x,i} P_i(x,x))^2} \cdot \frac{1}{\Phi_{\mathrm{RS\text{-}A}}^2}$$

Using this result, we upper bound the mixing time for systematic scan.

$$t_{\mathrm{mix}}(S, \epsilon) \leq \frac{2}{\Phi_{\mathrm{SS}}^2} \log\left(\frac{1}{\epsilon\pi_{\min}}\right) \leq \frac{8n^2}{(\min_{x,i} P_i(x,x))^2} \cdot \frac{1}{\Phi_{\mathrm{RS}}^2} \log\left(\frac{1}{\epsilon\pi_{\min}}\right)$$

This theorem results from the previous inequalities.

$$(1/2 - \epsilon)^2 \, t_{\mathrm{mix}}(S, \epsilon) \leq \frac{8n^2}{(\max_{x,i} P_i(x,x))^2} t_{\mathrm{mix}}(R, \epsilon)^2 \log\left(\frac{1}{\epsilon\pi_{\min}}\right)$$

□