[Reviews · NeurIPS 2016]

Reviewer 1

Summary

This paper is interested in comparing the mixing rates of Gibbs sampling using either systematic scan or random updates. The basic contributions are two: First, in Section 2, a set of cases where 1) systematic scan is polynomially faster than random updates. Together with a previously known case where it can be slower this contradicts a conjecture that the speeds of systematic and random updates are similar. Secondly, (In Theorem 1) a set of mild conditions under which the mixing times of systematic scan and random updates are not "too" different (roughly within squares of each other).

Qualitative Assessment

I think this paper addresses an important issue and makes valuable contributions, and thus should be published. I have a few concerns, hence my lower rating for the last question above (which I think could be addressed relatively easily, however). - The proofs are sketched out in quite an informal manner. I think this is fundamentally *OK* and even perhaps a positive thing. However, I think a bit more discussion needs to be given to how the arguments might be made more formal. For example, in Section 2.1, I think the proof is intended to hold only in the limit of M going to infinity. Please give a stament of what should hold in what limit-- this wasn't clear to me. Another example is the assumption that the bridge state has low mass in Section 2.2. This seems like a dubious assumption. Again this seems to be intended to hold only in the limit of a large model but exactly how this should work is not sketched out. (Also, if you are going to make this assumption, why not include it in Eq. 3?) Finally, the assumption that each of the islands mix rapidly within themselves again seems to weaken the strength of Section 2.2 All these assumptions are probably fine-- and improve the readability of the paper-- but I think more guidance needs to be given how one would make things more formal if desired. - I don't understand the experiments. Specifically, how are the mixing times calculated? This isn't discussed at all. I do not understand what was done to only consider states next to the bridge. - The argument at the end of Section 2.2 seems sufficient to compare f(n) to n*f(n) but it isn't clear why f(n) should itself be n. - What is |x| in P(x) on p. 4? Is this simply the length of x? - All the results seem to have the nature of adding/removing a factor of n, or squaring the mixing times. Is there anything fundamental about this? The abstract seems to lose something in only referring to this as "polynomial".

Confidence in this Review

2-Confident (read it all; understood it all reasonably well)


Reviewer 2

Summary

This paper presents several results concerning the dependency of the mixing time of a Gibbs sampler on its scan order. The scan order in a Gibbs sampler is typically random or in some systematic order. The paper gives two counterexample models where random and systematic scans differ by a polynomial factor in mixing time. Next, the authors prove under some conditions that different scan orders change the mixing time by at most a polynomial factor.

Qualitative Assessment

Overall, this is a nice paper, with clear writing and mostly sound arguments. It addresses an important problem, as Gibbs sampling or related methods are at the heart of many learning algorithms today. There are a couple of concerns: 1. The modification of the systematic scan order chain into a lazy Markov chain is necessary for the theory to go through. However, it is not clear how this relates to the original problem (a systematic scan order Gibbs sampler). Can the authors clarify if there is anything that can be said of the non-reversible systematic scan Gibbs sampler, given the result for a lazy version? 2. The authors introduce "mild conditions" (line 247) that the two probability quantities be at most polynomial in n. It is not clear to me that these are in fact mild assumptions. Why wouldn't it be reasonable to have entries in the transition matrix be exponential in n? Also, this feels like begging the question (assuming the terms in the bounds are polynomial in n to prove that the bounds are polynomial in n).

Confidence in this Review

2-Confident (read it all; understood it all reasonably well)


Reviewer 3

Summary

The authors study the convergence of the Gibbs sampler under various scanning strategies. They exhibit examples where the mixing times of random and systematic scan differ by more than a logarithmic factor and provide conditions under which these mixing times do not differ by more than a logarithmic factor.

Qualitative Assessment

Overall this is an interesting study of the Gibbs sampler in finite state-space. To the best of my knowledge, it presents novel counter examples and theoretical results, these results being obtained using conductance type techniques. I am a bit more negative about the relevance of this material for NIPS. First it is limited to finite state space whereas most Gibbs sampling type applications involve some continuous random variables. Second, there are however interesting applications of the Gibbs sampler to finite state-space (i.e. mixture models and LDA where the continuous parameters can be integrated out). It would have been interesting to study such a model in details. However, this is overall an interesting contribution.

Confidence in this Review

3-Expert (read the paper in detail, know the area, quite certain of my opinion)


Reviewer 4

Summary

The paper compares the mixing of random scan and systematic scan Gibbs samplers. First, following from a recent paper by Roberts and Rosenthal, the authors construct several examples which do not satisfy the commonly held belief that systematic scan is never more than a constant factor slower and a log factor faster than random scan. The authors then provide a result Theorem 1 which provides weaker bounds, which however they verify at least under some conditions. In fact the Theorem compares random scan to a lazy version of the systematic scan and shows that and obtains bounds in terms of various other quantities, like the minimum probability, or the minimum holding probability.

Qualitative Assessment

The paper is interesting and addresses a very interesting question and conjecture. It's well written, with only few typographical issues. I would have liked to see more rigorous arguments about the mixing times claimed in Section 2. For example, in the context Section 2.1, why is mixing equivalent to reaching the high mass state? The heuristic arguments provided are convincing, however if a simple rigorous argument could be given in the supplementary material it would be preferred. If a proof would be too long for the paper, then the authors could cite existing results which explain why the claim is correct. A couple of minor points: *systematic scan is a true Markov chain, but it is inhomogeneous while most results involving conductance etc are typically stated for homogeneous Markov chains. *in the statement of Theorem 1 "times with" seems out of place.

Confidence in this Review

3-Expert (read the paper in detail, know the area, quite certain of my opinion)


Reviewer 5

Summary

A study of how scan orders influence Mixing time in Gibbs sampling.

Qualitative Assessment

Tries to answer the question when does scan order matter in Gibbs sampling, in terms of mixing times. Specifically, the authors show specific cases where random scan is asymptotically better than Systematic scan and other examples of the inverse situation. They finally come up with a condition (Weaker than current conjecture) regarding the asymptotic equivalence of the the two scans. The paper is written excellently with clear examples and motivation. Gibbs sampling is probably one of the most popular approximate inference mechanism, so so results in to are always quite good for the research community. I am not entirely sure if the work here will have major practical applications (maybe the authors can elaborate on this a little) unless models fall into the structures suggested by the authors. But nevertheless, this is a strong paper.

Confidence in this Review

2-Confident (read it all; understood it all reasonably well)


Reviewer 6

Summary

MCMC is at the heart of many applications of modern machine learning and statistics. It is thus important to understand the computational and theoretical performance under various conditions. The present paper focused on examining systematic Gibbs sampling in comparison to random scan Gibbs. They do so first though the construction of several examples which challenge the dominant intuitions about mixing times, and develop theoretical bounds which are much wider than previously conjectured.

Qualitative Assessment

As noted by the authors, previous work (Roberts and Rosenthal) has shown the conjecture to be false in at least one direction. So the question becomes: Is it also false in the other directions? and Are there bounds to replace it? The bulk of the paper is devoted to analyzing various counter example inducing probability models and showing that mixing time can be heavily dependent on Gibbs scan order. These counter examples, though somewhat artificial, represent a useful contribution to the field, and will help practitioners understand the conditions under which ordering will likely matter in the construction of MCMC kernels. The primary theoretical result is Theorem 1, which puts polynomial bounds on the difference between random and (lazy) systematic scan Gibbs. This bound is to my knowledge novel and puts a useful worst case bound on the divergence of systematic and random scan mixing times. The technical need for the augmentation of the scan procedure does weaken the result somewhat. While the theoretical result was probably the most interesting aspect of the paper for me, its description and justification within the paper was cursory, with just the last 1.5 pages being devoted to it. The authors claim to outline a proof, but the discussion amounts to a lemma and theorem stated without proof, and then a claim that the result follows straightforwardly from them. Fortunately the required 5 pages of development were included in the supplemental; however, because of this, the paper does not stand alone without its supplemental. minor point: line 244 contains a typo. The authors may want to replace ". times with" with a ":"

Confidence in this Review

2-Confident (read it all; understood it all reasonably well)